# Deep Image Similarity Measurement Based on the Improved Triplet Network with Spatial Pyramid Pooling

**Xinpan Yuan [1], Qunfeng Liu [2], Jun Long [2],\* , Lei Hu [2] and Yulou Wang [2]**

[1] School of Computer, Hunan University of Technology, Zhuzhou 412000, China; xpyuan@hut.edu.cn
[2] School of Computer Science and Engineering, Central South University, Changsha 410083, China; qunfengliu@csu.edu.cn (Q.L.); hudalei@csu.edu.cn (L.H.); 174612284@csu.edu.cn (Y.W.)
\* Correspondence: jlong@csu.edu.cn; Tel.: +86-0731-8253-9926

**Abstract:** Image similarity measurement is a fundamental problem in the field of computer vision. It is widely used in image classification, object detection, image retrieval, and other fields, mostly through Siamese or triplet networks. These networks consist of two or three identical branches of convolutional neural network (CNN) and share their weights to obtain the high-level image feature representations so that similar images are mapped close to each other in the feature space, and dissimilar image pairs are mapped far from each other. Especially, the triplet network is known as the state-of-the-art method on image similarity measurement. However, the basic CNN can only handle fixed-size images. If we obtain a fixed size image via cutting or scaling, the information of the image will be lost and the recognition accuracy will be reduced. To solve the problem, this paper has proposed the triplet spatial pyramid pooling network (TSPP-Net) through combing the triplet convolution neural network with the spatial pyramid pooling. Additionally, we propose an improved triplet loss function, so that the network model can realize twice distance learning by only inputting three samples at one time. Through the theoretical analysis and experiments, it is proved that the TSPP-Net model and the improved triple loss function can improve the generalization ability and the accuracy of image similarity measurement algorithm.

**Keywords:** image similarity measurement; triplet network; spatial pyramid pooling; improved triplet loss function

## 1. Introduction

Through the characteristics of color, shape, texture, spatial structure, and semantic information, image similarity measurement aims to estimate whether a given pair of images are similar or not. It is one of the central problems in computer vision and pattern recognition, and is widely used in image search, image matching, image de-duplication, and other fields. A great performance of the image similarity measurement crucially depends on the feature representation and similarity measurement, which have been extensively studied by the multimedia researchers for decades. The most typical way of representing image features is the scale invariant feature transform (SIFT) [1]. These features are then encoded into image representations via various schemes, such as bag-of-words (BoW) [2–4]. Although a variety of techniques have been proposed, it remains one of the most challenging problems in current research, which is mainly due to the well-known "semantic gap" issue that exists between low-level image pixels captured by machines and high-level semantic concepts perceived by human. In order to solve this problem, we hope that the machine can deal with image similarity like human beings. Therefore, we need to find a better way to represent images and get their deep features.

For the past few years, deep learning models have been used extensively to solve the machine learning tasks. Especially, deep convolutional neural network (CNN) have a great performance in many computer vision tasks, including image classification [5,6], object detection [7,8], image retrieval [9,10], semantic segmentation [11,12], etc. With the deep architectures, semantic abstractions that are close to human cognition can be learned. A number of recent works show that CNN features trained on large and diverse datasets such as ImageNet [13] can be used to solve tasks for which they have not been trained. In addition, some complex convolutional neural network models have been proposed, such as Alexnet, VGGNet, GoogLetNet, ResNet, and so on. All of them had a great performance in the ImageNet Large Scale Visual Recognition Challenge (ILSVRC).

Recently, Siamese [14–16] and triplet networks [17,18] are methods to learn feature similarity by optimizing feature distance of image pairs. These networks use two or three identical branches CNN and share their weights to obtain the high-level image feature representation so that similar images are mapped close to each other in the feature space, and dissimilar image pairs are mapped far from each other. Therefore, these learning methods can obtain grate image features. Especially, the triplet network is known as the state-of-the-art method on image similarity measurement. Nowadays, it has been used in some application fields such like image retrieval and matching [19,20], face authentication [21–23], object recognition [24,25], and so on.

As we all know, deep convolutional neural network requires a fixed-size (e.g., 224_224) image as its input. In our real life, the size of the image often is not fixed, but varied. If we obtain a fixed size image via cutting or scaling, the information contained in the image will be lost and the recognition accuracy will be reduced. In order to remove the restriction that convolutional neural networks can only input a fixed-size image, Kaiming et al. proposed a spatial pyramid pooling (SPP) layer to remove the fixed-size constraint of the network [26]. Specifically, they add a SPP layer after the last convolutional layer. The SPP layer pools the features and generates fixed length outputs, which are then fed into the fully connected layers (or other classifiers). Therefore, the convolution neural network with a SPP layer can process any size images and obtain a fixed length output. In recent years, this method has been widely used in image classification [27,28] and object recognition [29–31].

In this paper, our main contributions are as follows:

1. We have proposed the triplet spatial pyramid pooling network (TSPP-Net) through combing the triplet convolution neural network with the spatial pyramid pooling, which can process any size images without cutting or scaling. It greatly improves the generalization ability of network and the accuracy of image similarity measurement.

2. We also have proposed an improved triple loss function, which can truly realize that the interclass distance is greater than the intraclass distance. It enables the triple network model to input three samples at a time and achieve two distance learning, which greatly improves the learning ability of the network. Additionally, the improved triple loss function can improve the sample utilization rate and accelerate the convergence speed of loss function, as well as reducing the iteration times of model training when the training dataset is limited.

The rest of the paper is organized as follows: Section 2 discusses the related works. Section 3 describes the improved triplet networks with spatial pyramid pooling in detail. Section 4 verifies the correctness and efficiency of the algorithm through experiments. Section 5 gives conclusions.

## 2. Related Work

This section will briefly introduce the Siamese network and triplet network, as well as the spatial pyramid pooling, so as to provide theoretical basis and technical support for the following research work. In Section 2.1, we will introduce the Siamese network and triplet network in detail, including their structures and their loss functions. In Section 2.2, we will describe the structure of the spatial pyramid pooling and how it can handle any size images.

### 2.1. Siamese Network and Triplet Network

The Siamese network [15,16] is a method of learning feature similarity by optimizing feature distance of image pairs, as shown in Figure 1. The Siamese network consists of two identical branches of CNN, which share their weights and parameters. Each branch of deep CNN does not have the last layer or classifier layer. The function $F(\cdot)$ represents the embedding features from each pair of images extracted by the Siamese network. This network model employs a pair of images $X_1$ and $X_2$ as the input and establishes a contrastive loss function ($L$), as show in Equation (1). The loss function tries to minimize squared Euclidean distance between the features of positive image pairs $D(X_1, X_2) = \|F(X_1) - F(X_2)\|_2^2$ and maximize it for negative pairs. The final goal is to minimize the loss function and find better network parameters through the training dataset:

$$L(X_1, X_2; m) = \frac{1}{2} \cdot Y \cdot D(X_1, X_2) + \frac{1}{2} \cdot (1 - Y) \cdot \{max(0, m - D(X_1, X_2))\} \tag{1}$$

where $Y$ is a binary label of the input images $X_1$ and $X_2$. $Y = 0$ indicates that the image pairs are dissimilar or negative and $Y = 1$ shows that the image pairs are similar or positive. The parameter $m$ is defined as the margin threshold between positive and negative.

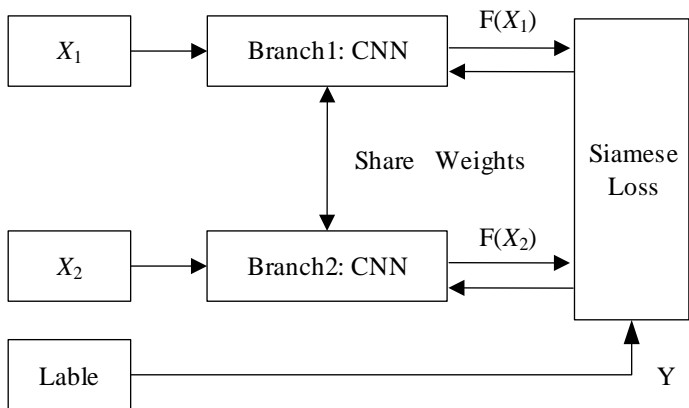

**Figure 1.** The structure of the Siamese network.

To learn a ranking function for image retrieval, Wang proposed the triple network model in 2014 [17]. Through the triplet constraints, triple network consists of three identical branches of CNN and share their weights, as shown in Figure 2. Additionally, google researchers developed a FaceNet on the basis of the triple network for face recognition and clustering [22]. The triplet network needs to input three images at the same time, namely anchor image($x^a$), positive image ($x^p$), and negative image($x^n$). The image pairs $x^a$ and $x^p$ are the same categories or similar images. Additionally, the image pairs $x^a$ and $x^n$ are the different categories or dissimilar images. Through the distance relationship of three images, the network model designs a triplet contrastive loss function ($L$), as show in Equation (2):

$$L(x^a, x^P, x^n; \alpha) = \frac{1}{N} \sum_i^N max\left\{D(x_i^a, x_i^p) - D(x_i^a, x_i^n) + \alpha, 0\right\} \tag{2}$$

where the parameter $\alpha$ represents the margin threshold between $D(x^a, x^p)$ and $D(x^a, x^n)$, $N$ denotes the number of triplet samples.

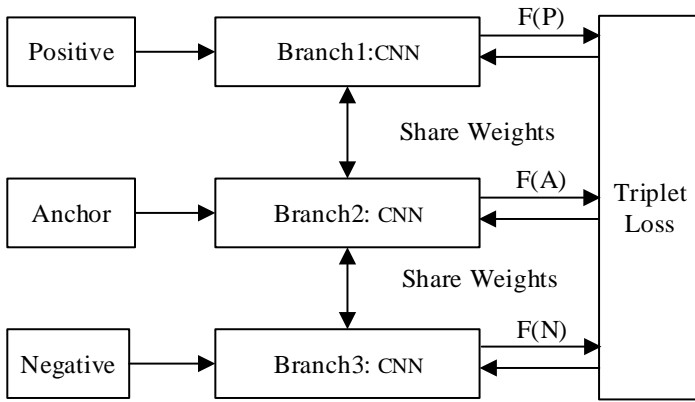

**Figure 2.** The structure of the triple network.

According to the triple network for face recognition developed by google, the learning target of the model is that similar image pairs are close to each other and dissimilar image pairs are far from each other, as shown in Figure 3. That is to say, the triplet network wants to minimize the distance between an anchor and a positive, both of which are similar images, and maximize the distance between the anchor and a negative of dissimilar images. From the distance relationship between images, the goal becomes that the distance between the anchor image and the positive image is less than that between the anchor sample and the negative sample. Therefore, there is a distance relationship as shown in Equation (3), for all the triplets in the training dataset:

$$D(x_i^a, x_i^p) + \alpha < D(x_i^a, x_i^n) \tag{3}$$

where $\alpha$ is a margin that is enforced between positive and negative pairs. The function $D(.)$ denotes the square of Euclidean distance.

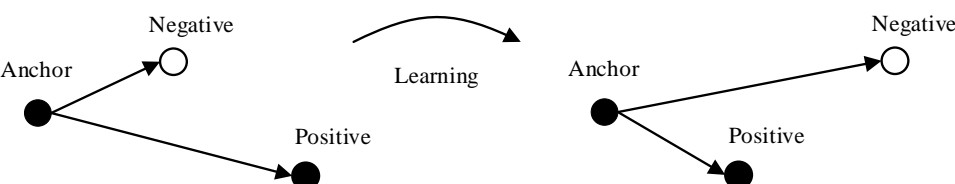

**Figure 3.** The learning target of the triplet network.

## 2.2. Spatial Pyramid Pooling

Deep convolutional neural network requires a fixed-size (e.g., 224 × 224) input image. In our real life, image size is not fixed, but varied. If we compulsively change the image size via cropping or warping, the information contained in the image will be lost. As a result, the accuracy of image classification and object recognition will be reduced. In convolutional neural network, convolutional layers do not require a fixed image size and can generate feature maps for any size of input. However, the fully-connected layers need to have fixed size length input due to their definition. That is to say, the fixed-size constraint of the network comes only from the fully-connected layers.

To solve this problem, a spatial pyramid pooling (SPP) layer was added on top of the last convolutional layer [26], as shown in Figure 4. The spatial pyramid pooling layer extracts the image features from the feature map through the 4 × 4, 2 × 2, and 1 × 1 square grid. Then, the SPP layer produces 21 = 16 + 4 + 1 different spatial bins and obtain a fixed size output by pooling each block. After the spatial pyramid pooling, any feature map can generate 5376-dimensional feature vector, where 5376 = 21 × 256. The SPP layer pools the features and generates fixed length outputs, which are then fed into the fully-connected layers. Therefore, the convolution neural network with spatial

pyramid pooling layer can deal with any size image and output a fixed size vector, without cropping or warping the image, which greatly improves the accuracy.

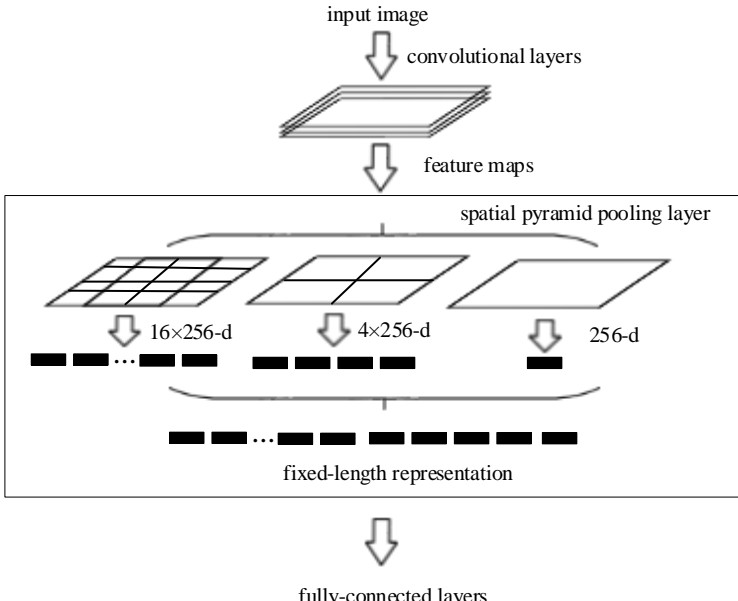

**Figure 4.** The spatial pyramid pooling layer.

## 3. The Improved Triplet Network with Spatial Pyramid Pooling

This section introduces, in detail, the improved triplet network and the image similarity measurement algorithm. Section 3.1 briefly describes the structure of the improved triplet network, as well as how to design the improved triplet loss function. Section 3.2 records the process of training the TSPP-Net model through the back propagation and mini-batch gradient descent. Section 3.2 also shows how to calculate the similarity between two images.

### 3.1. The Improved Triplet Network

Whether the Siamese network or the triplet network is composed of two or three branches of convolution neural network, and they can only process fixed-size images. To overcome this constraint, we propose the triplet spatial pyramid pooling network (TSPP-Net) via combining the triple network with the spatial pyramid pooling. The TSPP-Net also consist of three identical branches of CNN and share their weights. Additionally, for each CNN, a spatial pyramid pooling layer is added on top of the last convolutional layer to remove the constraints of inputting fixed-size images, as shown in Figure 5. Like the triple network [18,22], this model also needs to input three images, respectively anchor image($x^a$), positive image ($x^p$) and negative image($x^n$), where the image $x^a$ and the image $x^p$ are a similar image pairs whereas the image $x^a$ and the image $x^n$ are a dissimilar image pairs.

However, we find that the positive image $x^p$ and negative image $x^n$ are also a pair of dissimilar images. Thus, we have improved the original learning target and propose a new learning goal, as shown in Figure 6. The new goal can achieve twice-distance learning, including minimizing the distance between an anchor and a positive, maximizing the distance between an anchor and a negative, and maximizing the distance between a positive and a negative. After optimizing the learning objective, not only the distance between the anchor image and the positive image is larger than the distance the anchor image and the negative image, but also the distance between the anchor image and the positive image is larger than the distance between the positive image and the negative

sample, as showed in Equation (4). As a result, we can really realize that the inter-class distance is larger than the intra-class distance:

$$D(x_i^a, x_i^p) + \alpha < D(x_i^a, x_i^n), D(x_i^a, x_i^p) + \beta < D(x_i^p, x_i^n) \tag{4}$$

where $\alpha$ is the distance margin between $D(x^a, x^p)$ and $D(x^a, x^n)$. The parameter $\beta$ is the distance margin between $D(x^a, x^p)$ and $D(x^p, x^n)$.

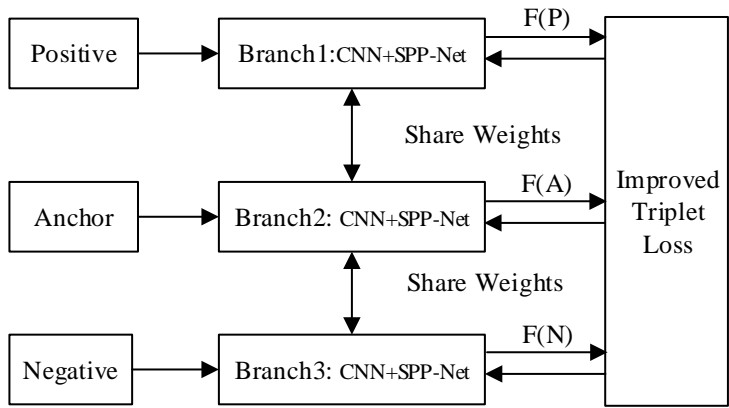

**Figure 5.** The construct of the triplet spatial pyramid pooling network.

Based on the original triple loss function and the new distance learning goals, we easily develop the improved triplet loss function, as showed in Equation (5). Compared with the original triple loss function, the improved triple loss function can realize twice distance learning only through a triple sample:

$$
\begin{aligned}
L(x^a, x^P, x^n, \alpha, \beta) \quad &= \frac{1}{N}\sum_i^N max\left\{D(x_i^a, x_i^p) - D(x_i^a, x_i^n) + \alpha, 0\right\} \\
&+ \frac{1}{N}\sum_i^N max\left\{D(x_i^p, x_i^a) - D(x_i^p, x_i^n) + \beta, 0\right\}
\end{aligned} \tag{5}
$$

where $N$ denotes the number of triplet samples.

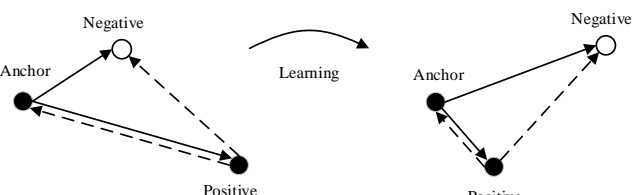

**Figure 6.** The improved learning goal of the triple network.

### 3.2. Image Similarity Measurement Algorithm

Based on a large number of image datasets, we train the TSPP-Net model and optimize the parameters by the back propagation and mini-batch gradient descent.

During the training process, the loss function becomes the Equation (6) and is defined as $L_m^*$:

$$
\begin{aligned}
L(x^a, x^P, x^n, \alpha, \beta) \quad &= \frac{1}{m}\sum_i^m max\left\{D(x_i^a, x_i^p) - D(x_i^a, x_i^n) + \alpha, 0\right\} \\
&+ \frac{1}{m}\sum_i^m max\left\{D(x_i^p, x_i^a) - D(x_i^p, x_i^n) + \beta, 0\right\}
\end{aligned} \tag{6}
$$

where $m$ represents the size of Mini-batch triplet images that selected randomly.

When $D(x^a, x^p) - D(x^a, x^n) + \alpha > 0$ and $D(x^p, x^a) - D(x^p, x^n) + \beta > 0$, we separately calculate the gradients for feature representation function $F(x_i^a)$, $F(x_i^p)$, and $F(x_i^n)$, then obtain the following formulas:

$$\frac{\partial L_m^*}{\partial F(x_i^a)} = \frac{1}{m}\sum_i^m \left[2F(x_i^a) - 4F(x_i^p) - 2F(x_i^n)\right] \tag{7}$$

$$\frac{\partial L_m^*}{\partial F(x_i^p)} = \frac{1}{m}\sum_i^m \left[-4F(x_i^a) + 2F(x_i^p) + 2F(x_i^n)\right] \tag{8}$$

$$\frac{\partial L_m^*}{\partial F(x_i^n)} = \frac{1}{m}\sum_i^m \left[2F(x_i^a) + 2F(x_i^p) - 4F(x_i^n)\right] \tag{9}$$

According to the chain rule of calculating the gradient for the compound function, we calculate the gradient of the loss function for the weights $W$, as shown in Equation (10):

$$\frac{\partial L_m^*}{\partial W} = \frac{\partial L_m^*}{\partial F(x_i^a)}\frac{\partial F(x_i^a)}{\partial W} + \frac{\partial L_m^*}{\partial F(x_i^p)}\frac{\partial F(x_i^p)}{\partial W} + \frac{\partial L_m^*}{\partial F(x_i^n)}\frac{\partial F(x_i^n)}{\partial W} \tag{10}$$

where:

$$\frac{\partial L_m^*}{\partial F(x_i^a)}\frac{\partial F(x_i^a)}{\partial W} = \frac{1}{m}\sum_i^m \left\{\left[2F(x_i^a) - 4F(x_i^p) - 2F(x_i^n)\right]\frac{\partial F(x_i^a)}{\partial W}\right\} \tag{11}$$

$$\frac{\partial L_m^*}{\partial F(x_i^p)}\frac{\partial F(x_i^p)}{\partial W} = \frac{1}{m}\sum_i^m \left\{\left[-4F(x_i^a) + 2F(x_i^p) + 2F(x_i^n)\right]\frac{\partial F(x_i^p)}{\partial W}\right\} \tag{12}$$

$$\frac{\partial L_m^*}{\partial F(x_i^n)}\frac{\partial F(x_i^n)}{\partial W} = \frac{1}{m}\sum_i^m \left\{\left[2F(x_i^a) + 2F(x_i^p) - 4F(x_i^n)\right]\frac{\partial F(x_i^n)}{\partial W}\right\} \tag{13}$$

Meanwhile, the gradient of the loss function for the biases $b$ is shown in Equation (14):

$$\frac{\partial L_m^*}{\partial b} = \frac{\partial L_m^*}{\partial F(x_i^a)}\frac{\partial F(x_i^a)}{\partial b} + \frac{\partial L_m^*}{\partial F(x_i^p)}\frac{\partial F(x_i^p)}{\partial b} + \frac{\partial L_m^*}{\partial F(x_i^n)}\frac{\partial F(x_i^n)}{\partial b} \tag{14}$$

where:

$$\frac{\partial L_m^*}{\partial F(x_i^a)}\frac{\partial F(x_i^a)}{\partial b} = \frac{1}{m}\sum_i^m \left\{\left[2F(x_i^a) - 4F(x_i^p) - 2F(x_i^n)\right]\frac{\partial F(x_i^a)}{\partial b}\right\} \tag{15}$$

$$\frac{\partial L_m^*}{\partial F(x_i^p)}\frac{\partial F(x_i^p)}{\partial b} = \frac{1}{m}\sum_i^m \left\{\left[-4F(x_i^a) + 2F(x_i^p) + 2F(x_i^n)\right]\frac{\partial F(x_i^p)}{\partial b}\right\} \tag{16}$$

$$\frac{\partial L_m^*}{\partial F(x_i^n)}\frac{\partial F(x_i^n)}{\partial b} = \frac{1}{m}\sum_i^m \left\{\left[2F(x_i^a) + 2F(x_i^p) - 4F(x_i^n)\right]\frac{\partial F(x_i^n)}{\partial b}\right\} \tag{17}$$

Therefore, the network updates its weights $W$ and biases $b$ as follows:

$$W_{l+1} = W_l - \eta \cdot \frac{\alpha L_m^*}{\partial W}\Big|_{W=W_l} \tag{18}$$

$$b_{l+1} = b_l - \eta \cdot \frac{\alpha L_m^*}{\partial b}\Big|_{b=b_l} \tag{19}$$

where $\eta$ is the learning rate, $l$ represents the layer.

To summarize, the process of training the TSPP-Net model is shown in Algorithm 1. After we already have trained the network model, we could choose a branch of CNN as the based network and

load the parameters. Then, we can obtain an image embedding or a feature vector from an image through the branch of CNN without the last layer.

Give two images $I_a$ and $I_b$, then obtain their corresponding feature vectors $A = [a_1, a_2, \ldots, a_n]$ and $B = [b_1, b_2, \ldots, b_n]$, where $n$ is the dimension of vector. Thus, the similarity between two images $S_1$ and $S_2$ can be defined as $Sim(A, B)$, as showed in Equation (20):

$$Sim(I_a, I_b) \approx Sim(A, B) = cos\langle A, B \rangle = \frac{\vec{A} \cdot \vec{B}}{\|A\| \cdot \|B\|} \tag{20}$$

The inputs of the Algorithm 1 include the training image dataset $I = \{I\}$, the total epoch number $E$, the iteration number $T$, the mini-bath size $m$, as well as the learning rate $\eta$. The outputs are the network parameters, including the weights $W = \{W\}$ and the biases $b = \{b\}$. Line 3 represents that the algorithm randomly generate the triplet samples: $X = \{(x_1^a, x_1^p, x_1^n), (x_2^a, x_2^p, x_2^n), \ldots, (x_n^a, x_n^p, x_n^n)\}$ through the image dataset $I$. Line 3 shows that the algorithm divide the triplet samples $X$ into the min-batch samples $M = \{M_1, M_2, \ldots\}$ and each of $M_i$ contains $m$ triplet samples, where $M_i = \{(x_{j+1}^a, x_{j+1}^p, x_{j+1}^n), (x_{j+2}^a, x_{j+2}^p, x_{j+2}^n), \ldots, (x_{j+m}^a, x_{j+m}^p, x_{j+m}^n)\}$. Line 9–12 describe that the algorithm calculates the gradient $\partial L_m^* / \partial w$ and $\partial L_m^* / \partial b$ by back propagation for all the min-batch samples $M_t$. Line 9 represents the algorithm updates the network parameters.

---

**Algorithm 1**: Training the TSPP-Net Model

---

Input:
　　　Training image dataset: $I = \{I\}$
　　　Epoch number: $T$
　　　Mini-Bath size: $m$
　　　Learning rate: $\eta$
Output:
　　　Network parameters: $W = \{W\}$, $b = \{b\}$

---

1:　　while $e < E$ do
2:　　　$e = e + 1$;
3:　　　Randomly generate the triplet samples: $X = \left\{(x_1^a, x_1^p, x_1^n), (x_2^a, x_2^p, x_2^n), \ldots, (x_n^a, x_n^p, x_n^n)\right\}$ through the image dataset $I$;
4:　　　Divide the triplet samples $X$ into the min-batch samples $M = \{M_1, M_2, \ldots\}$ and each of $M_i$ contains $m$ triplet samples, where $M_i = \{(x_{j+1}^a, x_{j+1}^p, x_{j+1}^n), (x_{j+2}^a, x_{j+2}^p, x_{j+2}^n), \ldots, (x_{j+m}^a, x_{j+m}^p, x_{j+m}^n)\}$;
5:　　　$T = size(M)$;
6:　　　while $t < T$ do
7:　　　　$t = t + 1$;
8:　　　Obtain the min-batch samples $M_t$ from $M$;
9:　　　for all the min-batch samples $M_t$:
10:　　　　According to the Equation (10), calculate the gradient $\partial L_m^* / \partial w$ by back propagation algorithm;
11:　　　　According to the Equation (14), calculate the gradient $\partial L_m^* / \partial b$ by back propagation algorithm;
12:　　　end for
13:　　　According to the Equation (18) and the Equation (19), update the parameters;
14:　　end while
15:　end while

---

## 4. The Experiment and Analysis

This section verifies the correctness and efficiency of the algorithm model through experiments. Section 4.1 briefly describes the experimental settings, including the experimental dataset and building the network model via the AlexNet and the spatial pyramid pooling. Section 4.2 introduces in detail the experimental results of the improved triplet network model, compared with the twin network and the triple network.

*4.1. Experimental Settings*

In order to avoid the effect of convolution network itself, we just use AlexNet as the base network, not VggNet, GoogLeNet and ResNet. AlexNet has three fully-connected layers (fc6, fc7, and fc8) and the last layer (fc8) was designed considering the number of classes on the image classification task. In this paper, we removed fc8 layer and used fc7 layer as the feature representations. Meanwhile, we added a spatial pyramid pooling (SPP) layer on top of the fully connected layer and obtain a fixed size embedding.

In this paper, we use the MNIST and Caltech101 to prove the correctness and validity of the network model. The details of image dataset are as follows:

**MNIST:** The MNIST dataset consists of handwritten digit images 0–9 and it is divided in 60,000 examples for the training set and 10,000 examples for testing. The official training set of 60,000 is divided into an actual training set of 50,000 examples and 10,000 validation examples. All digit images have been size-normalized and centered in a fixed size image of $28 \times 28$ pixels. In the original dataset each pixel of the image is represented by a value between 0 and 255, where 0 is black, 255 is white and anything in between is a different shade of grey.

**Caltech101:** The Caltech 101 dataset consists of a total of 9146 images, including 101 different object categories. Each category contains about 40–800 images. The size of each image is roughly $300 \times 200$ pixels.

According the Triple Network model [17,18], as well as the FaceNet model [22], we randomly generate a series of min-batch triplet pairs through the image dataset, each of which contains three images, respectively anchor image($x^a$), positive image ($x^p$) and negative image($x^n$). The image $x^a$ and the image $x^p$ have the same label, whereas the image $x^a$ and the image $x^n$ have different labels. We initialize the training parameters of the network model: the mini-batch size of 128, the total epoch number of 50, the learning rate of 0.0001. Additionally, to compare with the triple network model and align with the triple loss function, we have set a margin value $\alpha = \beta$ and modified the improved triplet loss function, as shown in the Equation (21):

$$
\begin{aligned}
L(x^a, x^P, x^n, \alpha, \beta) \quad &= \frac{1}{2m}\sum_{i}^{m} max\left\{D(x_i^a, x_i^p) - D(x_i^a, x_i^n) + \alpha, 0\right\} \\
&+ \frac{1}{2m}\sum_{i}^{m} max\left\{D(x_i^p, x_i^a) - D(x_i^p, x_i^n) + \alpha, 0\right\}
\end{aligned}
\tag{21}
$$

*4.2. Experimental Results*

In this paper, we mainly verify that the algorithm is correct and effective through the Improved Triple Loss Function and the TSPP-Net model. The results and analysis of the experiment are as follows:

4.2.1. Verifying the Improved Triple Loss Function

We use the MNIST dataset to verify the Siamese network, the triplet network, and the improved triplet network, respectively.

(1) Firstly, we select a series of margin values $\alpha$, such as 0.1, 0.2, 0.3, 0.5, 0.7, 0.8, 1.0, and so on. Then, we train these three network models separately and obtain the corresponding feature vectors. Finally, we classify these vectors classify via the KNN algorithm and analyze the accuracy of the three models along with the change of the margin values $\alpha$. The results are shown in Figure 7. When the margin value $\alpha = 0.1$, the classification accuracy of the Siamese, triplet, and improved triplet is 97.94%, 99.12% and 99.51%. This also effectively illustrates the improved triplet loss function, which can extract deeper image features. Its image embedding vector contains more abundant information, which has better advantages in the field of image similarity measurement. Obviously, the improved and optimized triple loss function network model has higher classification accuracy than the Siamese network and the original triple network.

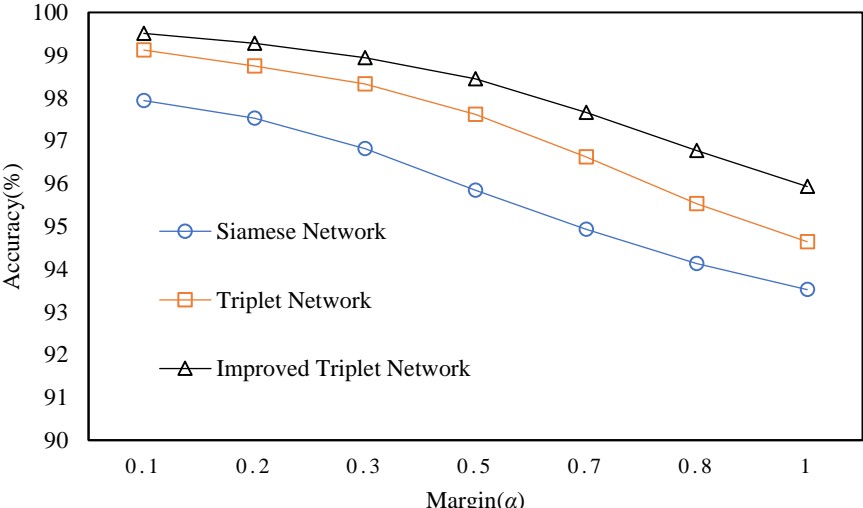

**Figure 7.** The classification accuracy of the three networks varies with a series of margin values.

(2) When the margin value $\alpha = 0.1$, we also have analyzed the convergence rate of the loss function and the classification accuracy of the three network models during the first 10 epoch training. The results are as shown in Figures 8 and 9. From the Figure 8, during the training process of the three network models through the MNIST dataset, when the epoch number reaches 10, the loss function of the triplet network and the improved triplet network has almost converged to 0. However, the loss function of Siamese network still fluctuates and does not converge. Therefore, the convergence rate of the loss function is: improved triplet network > triplet network > Siamese network. In Figure 9, the classification accuracy of the improved triplet network is the fastest and stable. The increasing rate of classification accuracy is: improved triplet network > triplet network > Siamese network. Based on the above analysis, it can be found that the improved loss function decreases fastest and converges fastest. When the loss function drops to the same value, the training network model has the least number of epochs and the highest sample utilization rate. In the case of relatively few training samples, improved triplet network can make full use of training samples and obtain some better network model parameters.

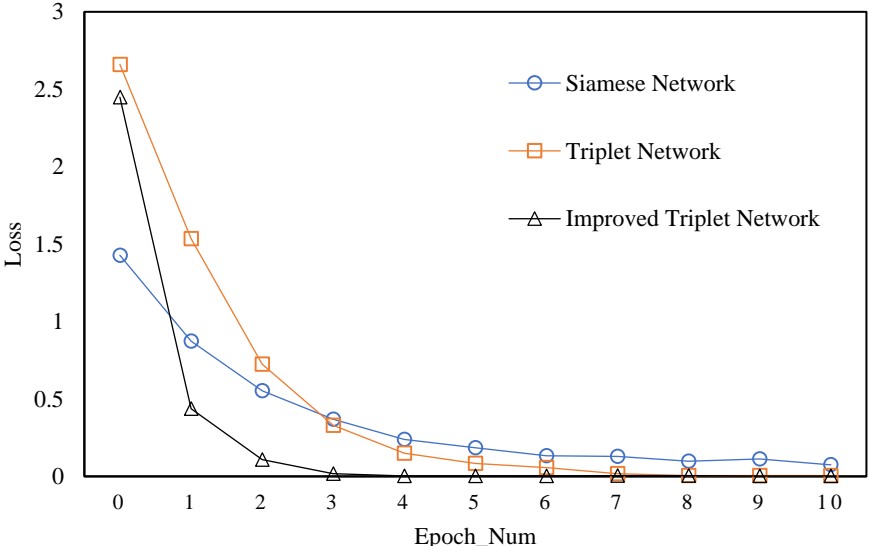

**Figure 8.** The convergence rate of the loss function varies with the epoch number.

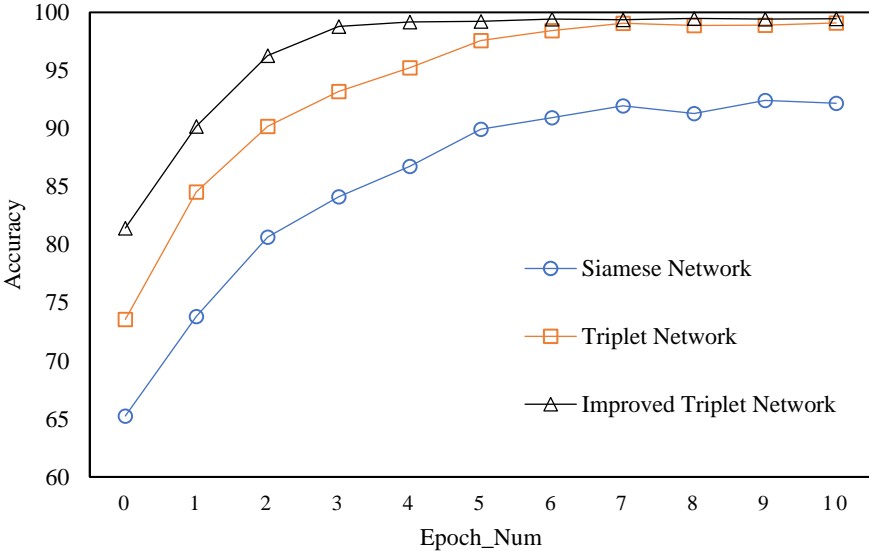

**Figure 9.** The classification accuracy of the three networks varies with the epoch number.

(3) When we have finished training the three network models, we load the parameters of the network model separately, then extract each image in the MNIST validation examples, and get the corresponding feature vectors. According to the Euclidean distance between the vectors, we project these feature vectors onto a two-dimensional space coordinate system, as shown in Figure 10. By analyzing the clustering results of the test dataset, we found that their performance is sorted: improved triplet network > triplet network > Siamese network. Therefore, compared with the Siamese Network and the original triplet network model, the improved triplet network has better learning ability. Similarly, it also shows that the improved triplet loss function can really realize that the inter-class distance is larger than the intra-class distance.

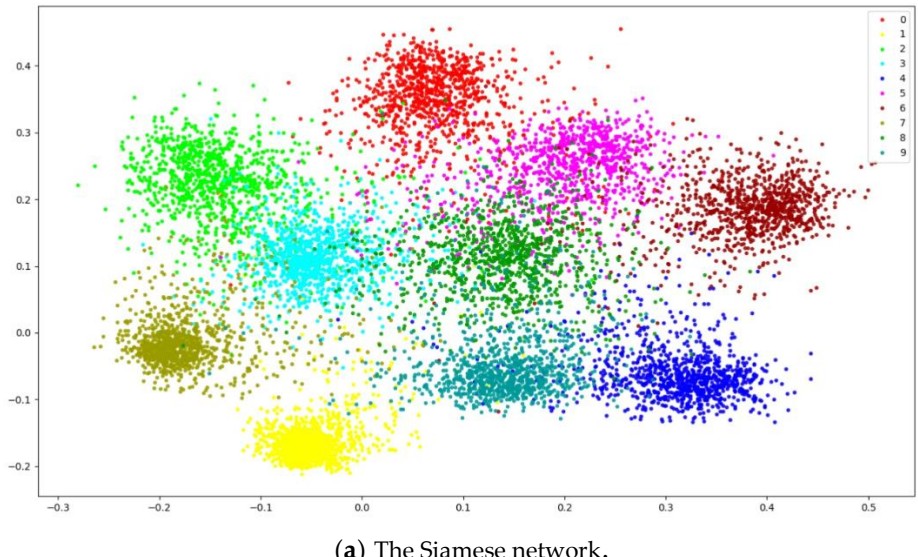

(**a**) The Siamese network.

**Figure 10.** *Cont.*

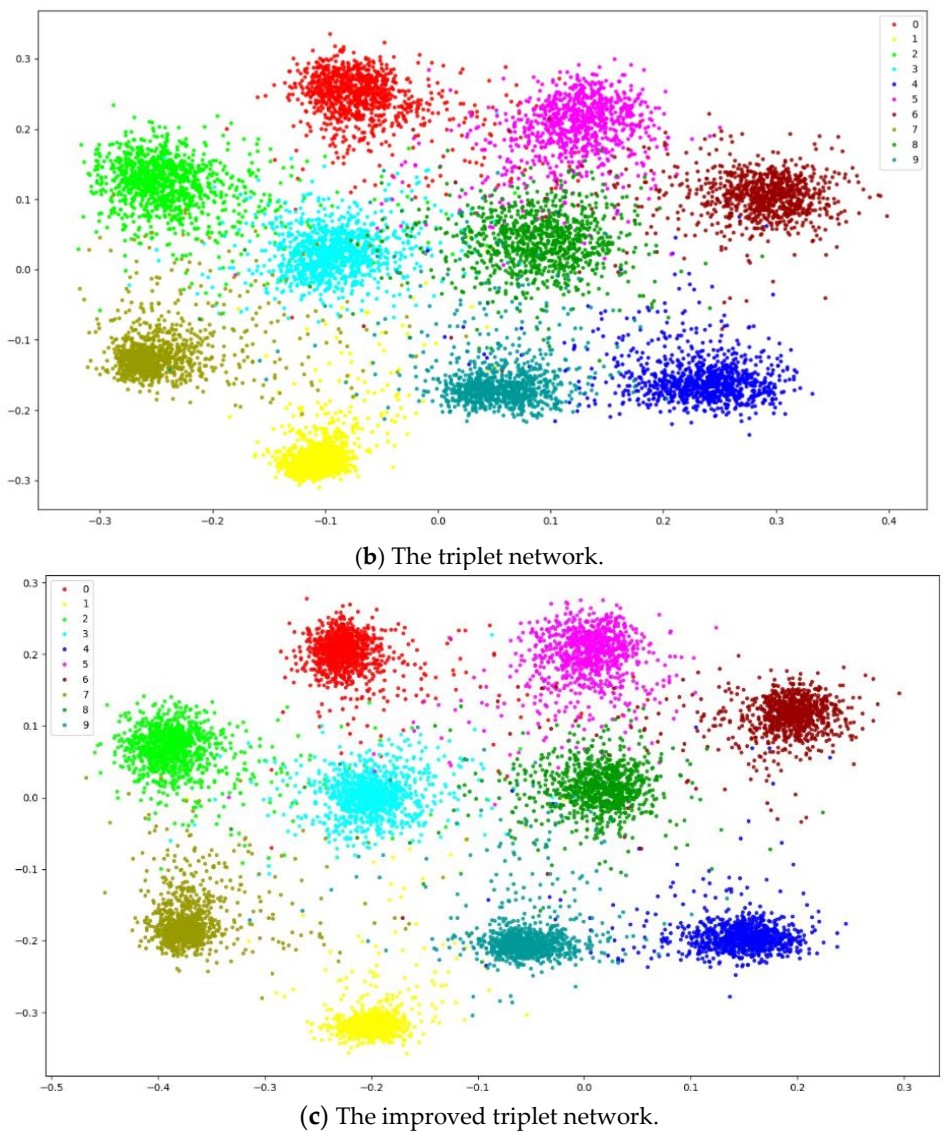

(**b**) The triplet network.

(**c**) The improved triplet network.

**Figure 10.** We project these feature vectors of the MNIST test dataset onto a two-dimensional space coordinate system, according to their Euclidean distance.

### 4.2.2. Verifying the TSPP-Net Model

According to the SPP-Net [26], we set the size of three grid blocks to be $4 \times 4$, $2 \times 2$, and $1 \times 1$, respectively. Then, if any size feature map is processed by the spatial pyramid pooling layer, 5376 features can be obtained, where $5376 = (16 + 4 + 1) \times 256$. At the same time, the SPP-Net provides two training methods: single size and multi-size. Since the Caltech 101 data is relatively complicated, we set a series of training parameters: the boundary value $\alpha$ of 0.5, the mini-batch size of 128, the total epoch number of 20,000, the learning rate of 0.0001. We selected 20 categories from the Caltech 101 dataset, and randomly selected 50 images from each category as the training data, and the rest of the images as the test data. We train Alexnet + SPP-Net (ASPP-Net), Siamese + SPP-Net (SSPP-Net), Triplet + SPP-Net (TSPP-Net) and Improved Triplet + SPP-Net (Improved TSPP-Net) through single-size and multi-size training. After training the network model, we use the model to extract image features from Caltech 101 dataset for image retrieval. We analyzed the precision rate, recall rate, and mean average precision (mAP).

(1) Single-Size Training

In the single-size training model, the image feature vectors extracted from the model are used for image retrieval, and the precision rate and the recall rate are shown in Figures 11 and 12. In the image retrieval results, as the parameter *k* increases gradually, the accuracy will decrease gradually, and the recall rate will increase gradually. When the values of *k* are the same, the accuracy rate is sorted: ASPP-Net < SSPP-Net < TSPP-Net < Improved TSPP-Net and the recall rate is sorted: ASPP-Net < SSPP-Net < TSPP-Net < Improved TSPP-Net. These show that the improved TSPP-Net model can extract better image features for image retrieval.

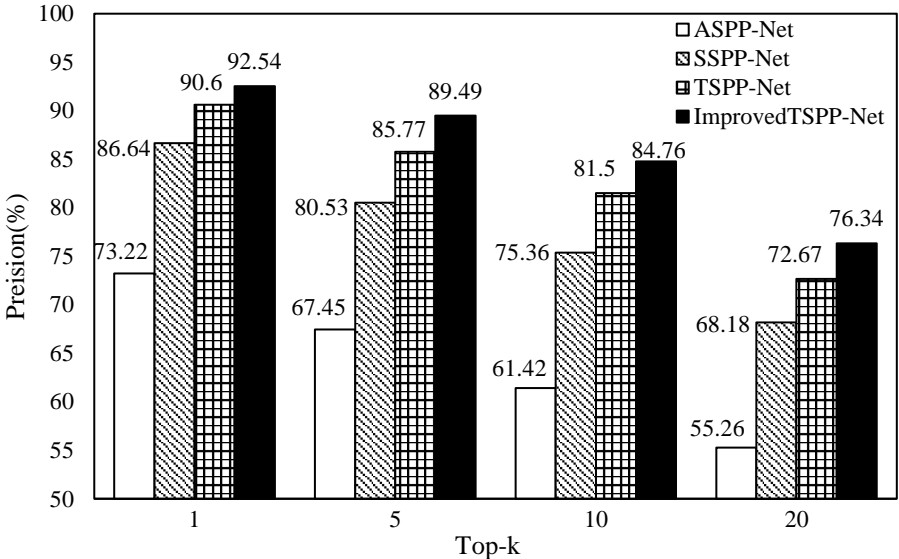

**Figure 11.** The precision rate of the single-size training.

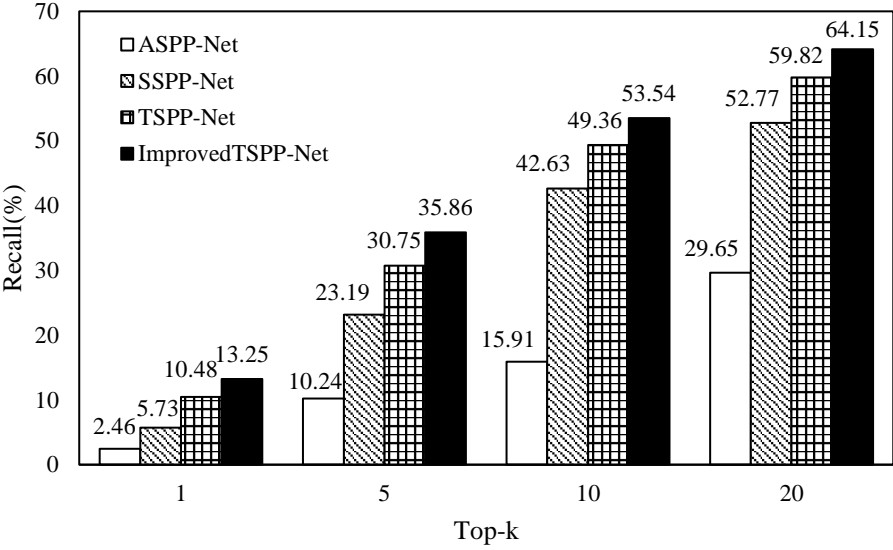

**Figure 12.** The recall rate of the single-size training.

(2) Multi-Size Training

According to the multi-size training method of the SPP-Net, the network model can be alternately and repeatedly trained through 224 × 224 and 180 × 180 images, where the 224 × 224 images are obtained by cutting the original image, while 180 × 180 size images are obtained by scaling the original image. Similarly, we extract image features through these trained network models for image retrieval and the results are shown in Figures 13 and 14.

Obviously, as the parameter *k* increases gradually, the accuracy will decrease gradually, and the recall rate will increase gradually. When the values of k are the same, the accuracy rate is: ASPP-Net < SSPP-Net < TSPP-Net < Improved TSPP-Net and the recall rate is: ASPP-Net < SSPP-Net < TSPP-Net < Improved TSPP-Net. Comparing the single-size training methods, the precision and recall rate of the multi-size training methods are slightly lower than the single-size training methods, but the difference is very small within an acceptable range. It shows that the multi-size training methods have less influence on the performance of the network model.

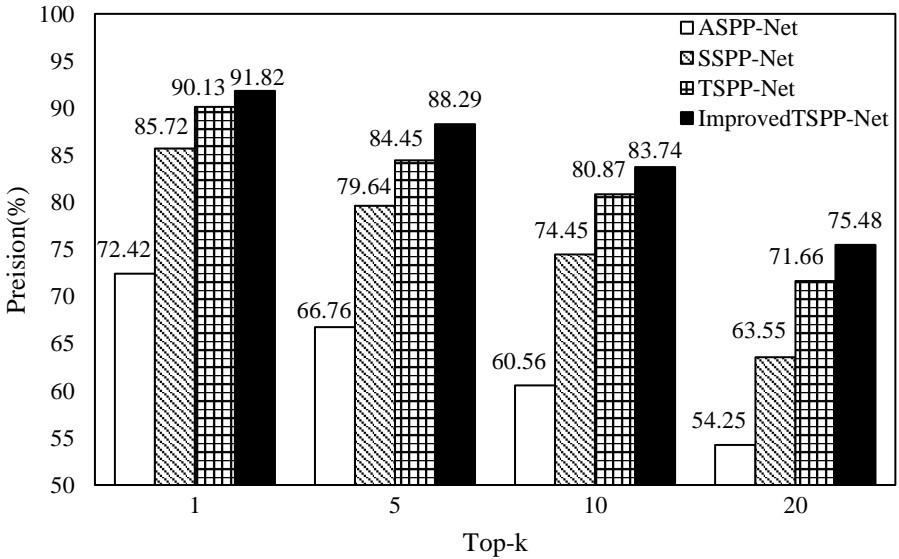

**Figure 13.** The precision rate of the multi-size training.

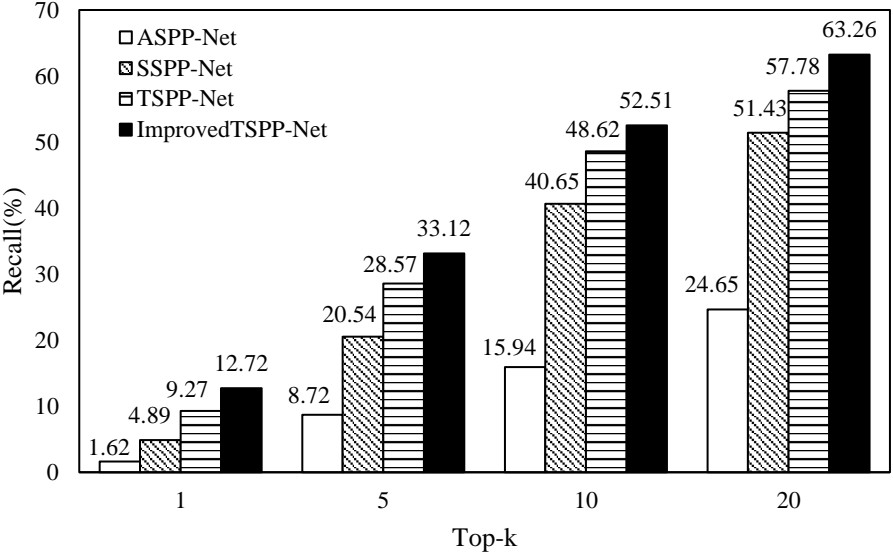

**Figure 14.** The recall rate of the multi-size training.

(3) Mean Average Precision

We obtain the image feature through these trained network models for image retrieval. the image retrieval results of mean average precision (mAP) are shown in Table 1. Under the two training modes of single-size and multi-size, the mean average precision is sorted: ASPP-Net < SSPP-Net < TSPP-Net < Improved TSPP-Net.

**Table 1.** Mean average precision.

| No. | Network Model | mAP (%) | |
|:---:|:---:|:---:|:---:|
| | | Single-Size | Multi-Size |
| 1 | ASPP-Net | 58.56 | 57.42 |
| 2 | SSPP-Net | 72.18 | 70.69 |
| 3 | TSPP-Net | 78.33 | 76.65 |
| 4 | Improved TSPP-Net | 81.24 | 79.35 |

## 5. Conclusions

In this paper, we have combined the triplet convolution neural network with the spatial pyramid pooling and proposed a TSPP-Net model, which effectively overcomes the limitation that the deep network model can only process fixed-size images. Additionally, we have proposed an improved triple loss function, which can truly realize that the interclass distance is greater than the intraclass distance and enable the triple network model to input three samples at a time to achieve two distance learning. According to theoretical analysis and experimental results, the TSPP-Net model and the improved triple loss function can improve the generalization ability of network and the accuracy of image similarity measurement algorithm, and have better advantages in the field of image similarity measurement. Furthermore, we use the TSPP-Net model for image similarity measurement algorithms and design image similarity detection systems. In the image similarity detection process for the National Natural Science Foundation project application, the system can quickly and accurately detect images with similar contents. The system is extremely stable and achieves our desired results.

However, there has always been a semantic gap in the field of image similarity measurement. Although the deep convolutional neural network can express the image information efficiently by simulating the thinking mode of the human brain, it narrows the gap between the machine and the human body in the semantic understanding, and better handles the semantic gap, but it is not really the semantics similarity. In the future, we can attempt to train the network model through multi-label learning or combining the semantic information contained in the image, so that we can really narrow the semantic gap.

**Author Contributions:** Conceptualization: X.Y. and Q.L.; formal analysis: J.L. and Q.L.; funding acquisition: J.L.; investigation: Q.L.; methodology: J.L. and Q.L.; project administration: X.Y. and Q.L.; software: X.Y. and Q.L.; supervision: X.Y.; validation: L.H; visualization: Y.W.; writing (original draft): J.L. and Q.L.; writing (review and editing): Q.L.

**Funding:** This work was supported in part by the National Natural Science Foundation of China (61402165, 61702560), the Key Research Program of Hunan Province (2016JC2018, 2018GK2052), the Natural Science Foundation of Hunan Province (2018JJ2099), and the Fundamental Research Funds for the Central Universities of Central South University (2017zzts510).

**Conflicts of Interest:** The authors declare no conflict of interest.

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
