# Peer review of "Deep Image Similarity Measurement Based on the Improved Triplet Network with Spatial Pyramid Pooling"

_information, doi:10.3390/info10040129_

Round 1

Reviewer 1 Report

The paper contains a lot of grammar errors, thus extensive editing of English language must be performed. 

The results are compared only with Siamese and Triplet networks. More valuable would be the comparison with the most efficient  approaches published in the literature, as well as for other datasets. 

Author Response

Dear Editors and Reviewers,

       We do appreciate your valuable suggestions on our paper entitled “Deep Image Similarity Measurement based on the Improved Triplet Network with Spatial Pyramid Pooling” (Manuscript ID: information- 474622). After receiving the suggestions, we have made great efforts to carefully revise the manuscript according to the comments and suggestions of reviewers. The responses to reviewers are as follows:

Minor issues:

Point 1: The paper contains a lot of grammar errors, thus extensive editing of English language must be performed.

Response 1: Thank you for your suggestions. In the revised paper, we have carefully revised the grammar errors, which are rewritten with the red color. We do hope that the revised paper can satisfy the strands of the Information Journal.

Point 2: The results are compared only with Siamese and Triplet networks. More valuable would be the comparison with the most efficient  approaches published in the literature, as well as for other datasets.

Response 2: Thank you for your advices. The paper proposed the triplet spatial pyramid pooling network (TSPP-Net) through combing the triplet convolution neural network with the spatial pyramid pooling. Siamese Network, Triplet Network and the TSPP-Net are composed of multiple branches of convolution neural network and have the same structure. Therefore, our experimental results are only compared with Siamese and Triplet networks. This paper also verifies the superiority and correctness of the model through the common image dataset MNIST and Catech101.

Thank you for your suggestions. We have revised this problem in the revised paper. We hope that the revised paper can satisfy the requirements of the journal.

Sincerely yours,

Authors: Xinpan Yuan, Qunfeng Liu, Jun Long, Lei Hu, Yulou Wang

Reviewer 2 Report

The article is quite interesting and the topic fits perfectly within the reach of the journal.

It is well structured, although I suggest the following changes to improve the quality of the article:

Line 84: before section 2.1, introduce the structure of section 2 and justify why it is so structured.

Line 222: before section 4.1, enter section 4 and its structure.

Section 5 should include at least one paragraph with future work.

There are some typos in the document.

Regarding the bibliography, it seems updated, but I would suggest incorporating some references from 2018 or 2019, since there is none.

Author Response

Dear Editors and Reviewers,

       We do appreciate your valuable suggestions on our paper entitled “Deep Image Similarity Measurement based on the Improved Triplet Network with Spatial Pyramid Pooling” (Manuscript ID: information- 474622). After receiving the suggestions, we have made great efforts to carefully revise the manuscript according to the comments and suggestions of reviewers. The responses to reviewers are as follows:

Minor issues:

Point 1: It is well structured, although I suggest the following changes to improve the quality of the article:

Line 84: before section 2.1, introduce the structure of section 2 and justify why it is so structured.

Line 222: before section 4.1, enter section 4 and its structure.

Response 1: Thank you for your suggestions. Before section 2.1, we wrote a paragraph to introduce the structure of section 2. The contents are as follows:

This Section will briefly introduce the Siamese Network and Triplet Network, as well as the spatial pyramid pooling, so as to provide theoretical basis and technical support for the following research work. In Section 2.1, we will introduce the Siamese Network and Triplet Network in detail, including their structures and their loss functions. In Section 2.2, we will describe the structure of the spatial pyramid pooling and how it can handle any size images.

We added a paragraph before section 3.1. The contents are as follows:

This section detailly introduces the Improved Triplet Network and the image similarity measurement algorithm. Section 3.1 briefly describes the structure of the Improved Triplet Network, as well as how to design the improved triplet loss function. Section 3.2 records the process of training the TSPP-Net model through the back propagation and Mini-batch Gradient Descent. Section 3.2 also shows how to calculate the similarity between two images.

We also added a paragraph before section 4.1 and the contents are as follows:

This section verifies the correctness and efficiency of the algorithm model through experiments. Section 4.1 briefly describes the experimental settings, including the experimental dataset and building the network model via the AlexNet and the spatial pyramid pooling. Section 4.2 introduces in detail the experimental results of the improved triplet network model, compared with the twin network and the triple network.

Point 2: Section 5 should include at least one paragraph with future work.

Response 2: Thank you for your advices. We added a paragraph to describe the future work in Section 5. The contents are as follows:

However, there has always been a semantic gap in the field of image similarity measurement. Although the deep convolutional neural network can express the image information efficiently by simulating the thinking mode of the human brain, it narrows the gap between the machine and the human body in the semantic understanding, and better handles the semantic gap, but it is not really the semantics similarity. In the future, we can try to train the network model through multi-label learning or combining the semantic information contained in the image, so that we can really narrow the semantic gap.

Point 3: Regarding the bibliography, it seems updated, but I would suggest incorporating some references from 2018 or 2019, since there is none.

Response 3: Thank you for your advices. We have revised and added some references from 2018 and 2019, which are rewritten with the red color. The references are as follows:

[1] Chen Z, Ding R, Chin T W, et al. Understanding the Impact of Label Granularity on CNN-based Image Classification[J]. 2019.

[2] Hui Z, Wang K, Tian Y, et al. MFR-CNN: Incorporating Multi-Scale Features and Global Information for Traffic Object Detection[J]. IEEE Transactions on Vehicular Technology, 2018, 67(9):8019-8030.

[3] Seddati O, Dupont S, Mahmoudi S, et al. Towards Good Practices for Image Retrieval Based on CNN Features[C]. IEEE International Conference on Computer Vision Workshops. 2018.

[4] Chen L C, Papandreou G, Kokkinos I, et al. DeepLab: Semantic Image Segmentation with Deep Convolutional Nets, Atrous Convolution, and Fully Connected CRFs[J]. IEEE Transactions on Pattern Analysis & Machine Intelligence, 2018, 40(4):834-848.

[5] Liu Z, Li X, Ping L, et al. Deep Learning Markov Random Field for Semantic Segmentation[J]. IEEE Trans Pattern Anal Mach Intell, 2018, PP(99):1-1.

[6] Huang C Q, Yang S M, Pan Y, et al. Object-Location-Aware Hashing for Multi-Label Image Retrieval via Automatic Mask Learning[J]. IEEE Transactions on Image Processing A Publication of the IEEE Signal Processing Society, 2018, 27(9):4490.

[7] Marin-Reyes P A, Bergamini L, Lorenzo-Navarro J, et al. Unsupervised Vehicle Re-identification Using Triplet Networks[C]. Conference on Computer Vision and Pattern Recognition Workshops. 2018.

[8] Taha A, Chen Y T, Misu T, et al. In Defense of the Triplet Loss for Visual Recognition[J]. 2019.

[9] Peng L, Liu X, Liu M, et al. SAR target recognition and posture estimation using spatial pyramid pooling within CNN[C]. Society of Photo-optical Instrumentation Engineers. 2018.

[10] Cao Y, Lu C, Lu X, et al. A Spatial Pyramid Pooling Convolutional Neural Network for Smoky Vehicle Detection[J]. 2018:9170-9175.

Thank you for your suggestions. We have revised this problem in the revised paper. We hope that the revised paper can satisfy the requirements of the journal.

Sincerely yours,

Authors: Xinpan Yuan, Qunfeng Liu, Jun Long, Lei Hu, Yulou Wang

Reviewer 3 Report

The assessment of image similarity is a notoriously difficult problem for automated solution. But it's an important problem and improvements in similarity measurement techniques would offer benefits in many fields. This paper examines the use of the CNN and discusses how it can be adapted to deal with images not all having the same size.

I found this paper very hard going. It is not accessible to someone who is not an expert in the field - it needs more introductory material. In particular, a brief overview of the approaches already adopted for image similarity measurement would be helpful. It's important to put the field of triplet networks into some kind of context. Despite what is said on lines 11-12, surely it is not the most popular technique for similarity measurement.

The paper really does need some careful and extensive editing - the English is not quite good enough to ensure that the meaning is always clear. For example, I found the start of Section 2 really muddled and the many typos (such as "google company") really do need attention. Avoid phrases such as "as we all know". The captions to the figures shouldn't contain extensive text explanations - that information should appear where the figure is referenced in the main body of the text.

The use of the large existing datasets for the evaluation of the proposed approach is sensible. But Section 4 is crying out for some proper statistical analysis - without that it's difficult to make firm conclusions. There are oddities here - for example the comments about convergence in the caption to Figure 8 need clarification. And I'm still unsure about the meaning of the "loss function decline speed". The results do look encouraging, but they really need to have some formal analysis.

The references need attention too. Many are incomplete and they need to be put into an appropriate and consistent format. Much information is missing here.

Overall, I feel that this work interesting, potentially useful and it is applicable to support the analysis of a wide range of image similarity problems. But the paper needs quite a lot of attention before it can be published.

Author Response

Response to Reviewer 3 Comments

Dear Editors and Reviewers,

       We do appreciate your valuable suggestions on our paper entitled “Deep Image Similarity Measurement based on the Improved Triplet Network with Spatial Pyramid Pooling” (Manuscript ID: information- 474622). After receiving the suggestions, we have made great efforts to carefully revise the manuscript according to the comments and suggestions of reviewers. The responses to reviewers are as follows:

Minor issues:

Point 1: I found this paper very hard going. It is not accessible to someone who is not an expert in the field - it needs more introductory material. In particular, a brief overview of the approaches already adopted for image similarity measurement would be helpful. It's important to put the field of triplet networks into some kind of context. Despite what is said on lines 11-12, surely it is not the most popular technique for similarity measurement.

Response 1: Thank you for your suggestions. We should say sorry to you, because of our paper being hard to understand.

We added a sentence in 16-17, as follows:

Especially, triplet Network is known as the state-of-the-art methods on image similarity measurement.

In the third paragraph of the Section 1 and Section 2.1, our paper detailly introduces the Siamese Network and Triplet Network, as well as their application in computer vision. To be more convinced, we have revised and added some references from 2018 and 2019, which are rewritten with the red color.

Point 2: The paper really does need some careful and extensive editing - the English is not quite good enough to ensure that the meaning is always clear. For example, I found the start of Section 2 really muddled and the many typos (such as "google company") really do need attention. Avoid phrases such as "as we all know". The captions to the figures shouldn't contain extensive text explanations - that information should appear where the figure is referenced in the main body of the text.

Response 2: Thank you for your advices. In the revised paper, we have carefully revised the grammar errors, which are rewritten with the red color. We revised all the captions of the figures and put the explanations in the text. We hope that the revised paper can satisfy the requirements of the journal.

Point 3: The use of the large existing datasets for the evaluation of the proposed approach is sensible. But Section 4 is crying out for some proper statistical analysis - without that it's difficult to make firm conclusions. There are oddities here - for example the comments about convergence in the caption to Figure 8 need clarification. And I'm still unsure about the meaning of the "loss function decline speed". The results do look encouraging, but they really need to have some formal analysis.

Response 3: Thank you for your advices.  First of all, we have changed the "loss function decline speed" to the "convergence rate of the loss function". Compared with the original triple loss function (equation 1), the improved triple loss function(equation 6) can realize twice distance learning only through a triple sample. Therefore, the improved triple loss function decreases faster and converges faster. At the same time, we analyzed them through some formulas and graphs (Figure 3 and Figure 6), in Section 3.2.

Besides, we added an experiment as follows

(3) When we have finished training the three network models, we load the parameters of the network model separately, then extract each image in the MNIST validation examples. and get the corresponding feature vectors. According to the Euclidean distance between the vectors, we project these feature vectors onto a two-dimensional space coordinate system, as shown in Figure 10. By analyzing the clustering results of these test samples, we found that their performance is: Improved Triplet Network> Triplet Network > Siamese Network. Therefore, compared with the Siamese Network and the original Triplet Network model, the improved Triplet Network has better learning ability. Similarly, it also shows that the improved triplet loss function can really realize that the inter-class distance is larger than the intra-class distance.

Point 4: The references need attention too. Many are incomplete and they need to be put into an appropriate and consistent format. Much information is missing here.

Response 4: Thank you for your advices. We have revised and added some references from 2018 and 2019, which are rewritten with the red color. The references are as follows:

[1] Chen Z, Ding R, Chin T W, et al. Understanding the Impact of Label Granularity on CNN-based Image Classification[J]. 2019.

[2] Hui Z, Wang K, Tian Y, et al. MFR-CNN: Incorporating Multi-Scale Features and Global Information for Traffic Object Detection[J]. IEEE Transactions on Vehicular Technology, 2018, 67(9):8019-8030.

[3] Seddati O, Dupont S, Mahmoudi S, et al. Towards Good Practices for Image Retrieval Based on CNN Features[C]. IEEE International Conference on Computer Vision Workshops. 2018.

[4] Chen L C, Papandreou G, Kokkinos I, et al. DeepLab: Semantic Image Segmentation with Deep Convolutional Nets, Atrous Convolution, and Fully Connected CRFs[J]. IEEE Transactions on Pattern Analysis & Machine Intelligence, 2018, 40(4):834-848.

[5] Liu Z, Li X, Ping L, et al. Deep Learning Markov Random Field for Semantic Segmentation[J]. IEEE Trans Pattern Anal Mach Intell, 2018, PP(99):1-1.

[6] Huang C Q, Yang S M, Pan Y, et al. Object-Location-Aware Hashing for Multi-Label Image Retrieval via Automatic Mask Learning[J]. IEEE Transactions on Image Processing A Publication of the IEEE Signal Processing Society, 2018, 27(9):4490.

[7] Marin-Reyes P A, Bergamini L, Lorenzo-Navarro J, et al. Unsupervised Vehicle Re-identification Using Triplet Networks[C]. Conference on Computer Vision and Pattern Recognition Workshops. 2018.

[8] Taha A, Chen Y T, Misu T, et al. In Defense of the Triplet Loss for Visual Recognition[J]. 2019.

[9] Peng L, Liu X, Liu M, et al. SAR target recognition and posture estimation using spatial pyramid pooling within CNN[C]. Society of Photo-optical Instrumentation Engineers. 2018.

[10] Cao Y, Lu C, Lu X, et al. A Spatial Pyramid Pooling Convolutional Neural Network for Smoky Vehicle Detection[J]. 2018:9170-9175.

Thank you for your suggestions. We have revised this problem in the revised paper. We hope that the revised paper can satisfy the requirements of the journal.

Sincerely yours,

Authors: Xinpan Yuan, Qunfeng Liu, Jun Long, Lei Hu, Yulou Wang

Round 2

Reviewer 1 Report

There are still a lot of grammmar errors, for example:

page 1: "branches convolutional neural network" => "branches of convolutional neural network"

 page 1,2: "state-of-the-art methods" => "state-of-the-art method"

page 2: "have a great"  => "has a great"

           "is often is not fixed" => "often is not fixed"

           "layer to to remove" => "layer to remove"

           "can process any size images and obtain a fixed length output" => "can proces images of any size and can obtain a fixed length output"

           "Recent years" => "In recent years"

page 3: "pair images" => "pair of images"

page 4: "Triplet Network want to " => "Triplet Network wants to "

             "produce " => "produces"

            "Maximize pooling each block and obtain a fixed size output." ????

etc.

Therefore, the paper should be carefully verified and all errors should be corrected.

Author Response

Dear Editors and Reviewers,

    We do appreciate your valuable suggestions on our paper entitled “Deep Image Similarity Measurement based on the Improved Triplet Network with Spatial Pyramid Pooling” (Manuscript ID: information- 474622). After receiving the suggestions, we have made great efforts to carefully revise the manuscript according to the comments and suggestions of reviewers. The responses to reviewers are as follows:

Minor issues:

Point:

There are still a lot of grammmar errors, for example:

page 1: "branches convolutional neural network" => "branches of convolutional neural network"

 page 1,2: "state-of-the-art methods" => "state-of-the-art method"

page 2: "have a great"  => "has a great"

           "is often is not fixed" => "often is not fixed"

           "layer to to remove" => "layer to remove"

           "can process any size images and obtain a fixed length output" => "can proces images of any size and can obtain a fixed length output"

           "Recent years" => "In recent years"

page 3: "pair images" => "pair of images"

page 4: "Triplet Network want to " => "Triplet Network wants to "

             "produce " => "produces"

            "Maximize pooling each block and obtain a fixed size output." ????

etc.

Therefore, the paper should be carefully verified and all errors should be corrected.

Response:

Thank you for your suggestions. In the revised paper,we have carefully revised the grammar errors, which are rewritten with the green color.

Thank you for your suggestions. We have revised this problem in the revised paper. We hope that the revised paper can satisfy the requirements of the journal.

Sincerely yours,

Authors: Xinpan Yuan, Qunfeng Liu, Jun Long, Lei Hu, Yulou Wang

Reviewer 3 Report

Much has improved in this version. As requested, the start of Section 2 has been modified to include more background information. And it's good to see the graphically-presented results from the varous classifiers. I'm still a little concerned about the English - there are still *many* typos and muddled grammar. If this can be addressed in the production process, then I'd be happy. There's still no formal statistical analysis too. I really don't know whether or not this matters!

Author Response

Dear Editors and Reviewers,

       We do appreciate your valuable suggestions on our paper entitled “Deep Image Similarity Measurement based on the Improved Triplet Network with Spatial Pyramid Pooling” (Manuscript ID: information- 474622). After receiving the suggestions, we have made great efforts to carefully revise the manuscript according to the comments and suggestions of reviewers. The responses to reviewers are as follows:

Minor issues:

Point:

Much has improved in this version. As requested, the start of Section 2 has been modified to include more background information. And it's good to see the graphically-presented results from the varous classifiers. I'm still a little concerned about the English - there are still *many* typos and muddled grammar. If this can be addressed in the production process, then I'd be happy. There's still no formal statistical analysis too. I really don't know whether or not this matters!

Response:

Thank you for your suggestions. In the revised paper,we have carefully revised the grammar errors, which are rewritten with the green color.

Thank you for your suggestions. We have revised this problem in the revised paper. We hope that the revised paper can satisfy the requirements of the journal.

Sincerely yours,

Authors: Xinpan Yuan, Qunfeng Liu, Jun Long, Lei Hu, Yulou Wang
